# Antibiotics Knowledge, Attitudes and Behaviours among the Population Living in Greece and Turkey

**DOI:** 10.3390/antibiotics12081279

**Published:** 2023-08-03

**Authors:** Timo J. Lajunen, Mark J. M. Sullman, Buket Baddal, Burcu Tekeş, Menelaos Apostolou

**Affiliations:** 1Department of Psychology, Norwegian University of Science and Technology, NO-7491 Trondheim, Norway; 2Department of Social Sciences, School of Humanities and Social Sciences, University of Nicosia, Nicosia CY-1700, Cyprus; sullman.m@unic.ac.cy (M.J.M.S.); apostolou.m@unic.ac.cy (M.A.); 3Department of Medical Microbiology and Clinical Microbiology, Faculty of Medicine, Near East University, Nicosia 99138, Cyprus; buket.baddal@neu.edu.tr; 4Department of Psychology, Başkent University, Ankara 06790, Turkey; burcutekes@baskent.edu.tr

**Keywords:** antibiotics, antimicrobial resistance, attitude, knowledge, Greece, Turkey

## Abstract

Antimicrobial resistance is one of the largest threats to public health worldwide. As the inappropriate use of antibiotics is one of the leading causes of antibiotic resistance, it is important to have an understanding of the public’s knowledge, attitudes and behaviours towards antibiotics and antimicrobial resistance. The present study investigated the knowledge, attitudes and behaviours towards antibiotics among the public living in Greece and Turkey using an online cross-sectional survey, with social media advertising (e.g., Facebook) and snowball sampling. In total, 709 individuals completed the survey (Greece *n* = 309, Turkey *n* = 400), with an average age of 34.2 (SD = 13.1) and 40.5 (SD = 14.7), respectively. In Greece, 49.4% of the participants were female, and in Turkey, this figure was 62.4%. The Greek respondents reported that obtaining antibiotics without a prescription was easier (52.6% reported “easy or very easy”) than the Turkish (35.6% reported “easy or very easy”) respondents did. This study reveals that Greek citizens were more educated and knowledgeable about antibiotics (58.5% of Greeks and 44.2% of Turks identified antibiotics correctly), their effects (20.9% of Greeks and 26.3% of Turks agreed with wrong statements about antibiotics) and the risks of antibiotic resistance, compared to those from Turkey. On the other hand, the Greek respondents were more prone to use leftover antibiotics or to give them to someone else later (*p* < 0.001). The findings of this study indicate that Greece and Turkey, both countries with high rates of antibiotic usage, exhibit distinct variations in their knowledge, attitudes, and perceptions concerning antibiotic use and antibiotic resistance. Effective countermeasures such as public campaigns should be targeted according to the population and those areas of knowledge, attitudes and behaviours in which the main shortcomings lie.

## 1. Introduction

Antimicrobial resistance (AMR) is a substantial cause of morbidity and mortality across the globe. In 2019, AMR led to the deaths of at least 1.27 million people and was associated with nearly five million deaths worldwide [1]. Furthermore, in the Europe Region the countries with the largest defined daily doses (DDD) of antibiotics per 1000 inhabitants were Greece and Turkey [2]. Although the WHO report was based upon 2015 data, the 2018 estimates for Turkey (38.18 DDD) and Greece (45.9 DDD) were also extremely high [3]. Moreover, in 2020, Greece had the highest consumption of antibiotics (28.1 DDD) in the Organisation for Economic Cooperation and Development (OECD), while Turkey (24.4 DDD) had the third highest [4].

The unnecessary and inappropriate use of antibiotics is the main factor underlying the development of AMR [5]. Therefore, a thorough understanding of the general public’s knowledge, attitudes and practices regarding antibiotic use and AMR is essential for the generation of appropriate interventions [6]. Several recent studies have investigated the knowledge, attitudes and practices (KAP) of the general public in many different European countries [7,8,9,10], but not a large number in Turkey or Greece.

In Turkey, several studies have investigated the attitudes, knowledge and behaviours towards antibiotics [11,12,13,14], but only one surveyed more than one city [13]. A study of 945 Turkish university students and their friends and families found that over half (55.8%) reported having used antibiotics in the past 12 months, and over a third (34.2%) had self-medicated without a prescription [13]. Furthermore, although 80.4% knew that antibiotics are used to treat bacterial infections, 51.4% thought they could also treat viral diseases. In addition, a belief in the effectiveness of antibiotics for treating different symptoms was the most influential factor in determining the frequency of antibiotic use. Conversely, negative attitudes towards antibiotics, having good health and possessing adequate knowledge about antibiotics were associated with less frequent use.

A study of 1057 Bursa residents found that many believed antibiotics could be used to treat bacterial infections (58.3%), viral infections (75.5%), as well as coughs, colds and the flu (60.4%) [11]. In addition, 64.5% believed it was correct to stop taking antibiotics when symptoms improved and that it was healthier to take fewer antibiotics than were prescribed (40.0%). They also found a high prevalence of patients pressuring physicians into prescribing antibiotics by complaining of unrealistic health symptoms, particularly among those with a low level of education. Insufficient public knowledge about antibiotics and their misuse, overtreatment expectancy, and pressuring physicians to prescribe antibiotics were identified as essential factors in antibiotic abuse.

A study in Ankara (*n* = 322) found that 64.3% of the participants reported self-medicating with antibiotics, with a cold and fever being the leading reasons for self-medicating [14]. The majority of the participants (64%) reported that they demand a prescription for antibiotics from the doctor, 64.9% reported keeping antibiotics at home and 87% stated they could buy antibiotics without a prescription. In another Ankara study (*n* = 1044), they found that 63.7% of the participants had used antibiotics in the last 12 months, and 19.4% indicated that they stopped taking antibiotics when their symptoms improved [12]. In total, 27.2% of people used antibiotics without visiting a physician, 24.3% demanded a prescription for antibiotics when visiting their physician and 83.5% of them reported that the physician fulfilled their request. They also found that the appropriate use of antibiotics was higher among those who were informed by physicians about the possible harm from their inappropriate use and those having a university degree.

Although Greece remains one of the largest consumers of antibiotics, there have been surprisingly few studies investigating the attitudes, knowledge and behaviours about antibiotics in the last decade. However, during the COVID-19 pandemic, researchers investigated parental (*n* = 412) knowledge, attitudes and behaviours towards antibiotic use in children with upper respiratory tract infections (URTIs) [15]. Although most parents (86%) correctly believed that viruses primarily caused cold or flu symptoms, 26.9% thought that antibiotics might be necessary to prevent complications.

Similarly, the majority of parents (85%) were aware that COVID-19 originated from a viral source, but approximately half of them (49.8%) were unsure about the necessity of antibiotics in its treatment. Furthermore, the study revealed that all demographic characteristics, except for gender, exhibited a noteworthy impact on parents’ knowledge, attitudes and practices concerning antibiotic use for upper respiratory tract infections (URTIs) and COVID-19.

The current study examined the knowledge, attitudes and practices related to antibiotics within the general public of Turkey and Greece. The study used a cross-sectional survey, Facebook advertising and snowball sampling to obtain samples from the general public of both countries. These findings hold the potential to steer future interventions aimed at enhancing awareness and curbing unnecessary antibiotic usage in both nations.

## 2. Materials and Methods

### 2.1. Participants and Data Collection

Sample size estimation using G*Power 3.1 application (effect size 0.5, power 0.95) indicated that the minimum sample size for sample comparisons was 184. The sample sizes were 309 for Greece and 400 for Turkey. The inclusion criteria for participants were age ≥ 18 and being a resident in Greece or Turkey. Since the questionnaire was presented in Turkish in Turkey and in Greek in Greece, only individuals with knowledge of these languages could participate. The questionnaire included a question about mother tongue: all Greek participants reported their mother tongue being Greek and all Turkish participants reported their mother tongue being Turkish.

The questionnaires were translated into the local languages using native speakers of English and the local languages. These were then back translated using a second native speaker of both languages.

The design of the study followed the protocol proposed in the Checklist for Reporting Results of Internet E-Surveys (CHERRIES) with minor adjustments for the study. To collect data, an online questionnaire was administered through Google Forms, and recruitment was carried out using advertising on social media platforms, such as Facebook, as well as employing snowball sampling techniques. In addition to social media advertising, friends, family members and colleagues were emailed and asked to forward the questionnaire link to people in their network. Participants were all provided with comprehensive information about the study’s background, objectives, voluntary participation, as well as an assurance of anonymity and data confidentiality. All participants provided informed consent before engaging in the study, and the research protocol received approval from the Social Science Ethics Research Board (SSERB) at the University of Nicosia (Approval Number: SSERB 00137).

### 2.2. Questionnaire

The questionnaire comprised six sections. Section 1 focused on participants’ antibiotic usage history within the past 12 months [9,10,16]. It included questions about the frequency of antibiotic consumption (e.g., once, 2–5 times, more than 5 times), adherence to doctor’s advice, instances of antibiotic usage without a prescription and the perceived ease of obtaining antibiotics without a prescription. Section 2 involved participants identifying whether five specific drugs were antibiotics or not, with response options of “Yes”, “No” or “Don’t Know”. Section 3 consisted of two parts: the first assessed participants’ knowledge regarding antibiotics (e.g., Antibiotics kill viruses), and the second addressed antibiotic resistance (e.g., Bacteria may become resistant to antibiotics). Both knowledge and resistance were assessed through 13 statements. The respondents evaluated the correctness of these statements by choosing one of three alternatives (“Yes”, “No” or “Don’t Know”). In Section 4, participants were presented with 10 questions that measured their personal attitudes towards antibiotics (e.g., I prefer to use antibiotics only when necessary). These questions were sourced from [9,10]. The participants answered the statements using a 5-point Likert scale ranging from “Strongly agree” to “Strongly disagree“. After recoding the negatively keyed items (1, 4, 6, 7), a sum score for “positive attitude to antibiotics use” was calculated. Alpha reliability coefficient of the sum scale was 0.67. Section 5 consisted of 12 questions sourced from the Beliefs about Medicines Questionnaire [14], which measured general attitudes towards medications. Participants responded to these questions using a 5-point Likert scale (“Strongly agree” to “Strongly Disagree”). Since these questions did not show the required psychometric quality, they were excluded from the final analyses. The final section requested demographic information, including sex, age, marital status, smoking status, self-perceived general health, household income, nationality, and native language.

### 2.3. Statistical Analyses

The study utilized IBM SPSS Statistics for Windows, version 25 (IBM, Tokyo, Japan), to conduct all analyses. To compare the knowledge, attitudes, and practices of participants from the two countries, independent samples *t*-tests, Chi-square tests and Mann–Whitney U tests were employed. A significance level of *p* < 0.05 was used to determine statistical significance.

## 3. Results

### 3.1. Population Characteristics

The data comprise responses from 309 participants from Greece (mean age = 34.2, SD = 13.1) and 400 participants from Turkey (mean age = 40.5, SD = 14.7). The sample characteristics are displayed in Table 1, which indicate that the Turkish sample tended to be slightly older (*p* < 0.001) and had a higher proportion of females, more married individuals and a higher level of education compared to the Greek sample. Additionally, participants were questioned about their smoking habits, physical fitness, general health and whether they had a health professional background. There were no significant differences between the two countries with regards to the proportion that had a health professional background, smoking status and whether they reported themselves to be in good health. However, more people in the Greek sample considered themselves to be fit.

### 3.2. Previous Antibiotic Use

Participants were then asked about their recent experiences with antibiotics (Table 2). More than 40% (Greece = 44.8%, Turkey = 42.4%) of the respondents had used antibiotics in the last year. Table 2 also shows that among the users of antibiotics, most reported having used only one course of antibiotics over the last year (Greece = 31.2%, Turkey = 24.6%). The vast majority of participants (Greece = 93.5%, Turkey = 92.0%) reported that they followed the doctor’s orders when using antibiotics, with just over 5% (Greece = 5.5%, Turkey = 5.8%) reporting that they did not. The respondents were also asked whether they had requested a prescription for antibiotics, although unnecessary. There were no significant differences between Greece and Turkey, with most reporting that they had not requested antibiotics (Greece = 88.3%, Turkey = 89.2%). Interestingly, only 15.6% of Greek citizens thought it was either difficult or extremely difficult to obtain antibiotics without a prescription, while this figure was 40.6% in Turkey, indicating a much more strict enforcement of the rules in Turkey (*p* < 0.001).

### 3.3. Recognition of Antibiotics

Participants were also asked whether they were able to recognise which medications were antibiotics (Table 3). Table 3 presents the distribution of answers among the Greek and Turkish respondents, with their answers compared using Mann–Whitney U tests. The most correctly recognized antibiotics were amoxicillin in Greece (70.1%) and penicillin in Turkey (60.9%). Penicillin was the only medication with equal recognition rates in both samples, which is not surprising considering this was the first antibiotic to be discovered and thus has a long history of use. Considerably more Greeks recognised tetracycline (Greece = 44.5%, Turkey = 26.6%, *p* < 0.001) and amoxicillin as antibiotics (Greece = 70.1%, Turkey = 45.1%, *p* < 0.001). There were more Turkish individuals (12.8%) than Greek individuals (6.8%) who thought ibuprofen was an antibiotic and more Greek citizens (10.1%) thought paracetamol was an antibiotic than did Turkish citizens (6.8%). In summary, the country comparisons (Mann–Whitney statistics) show there were significant intercountry differences for all but penicillin. In general, Greeks seemed to be better at recognizing antibiotics than Turks. Without the knowledge of which pharmaceuticals are antibiotics and which are not, patients cannot be expected to use the medications as prescribed.

### 3.4. Antibiotic Use and Effects

Table 4 shows the differences between the Greek and Turkish samples, in terms of knowledge about antibiotic use and its effects. Statistical differences between the Greek and Turkish citizens were found in 8 of the 13 statements. Greek individuals were more likely than Turkish individuals to believe that antibiotics kill viruses, although in both samples the clear majority gave the right answer. Also, more Turkish respondents knew that antibiotics are effective against bacteria than did Greek respondents. In contrast, more Turkish people thought that antibiotics are effective against colds than did Greeks, which might indicate that the viral nature of the common cold is not completely clear to the respondents. As Table 4 shows, more Turkish respondents thought that penicillin was another word for antibiotics than did Greeks. Turkish individuals seemed less knowledgeable about the side effects (diarrhoea and vaginal fungus) than Greeks were. In addition, Turkish people saw less risk than Greek people in taking antibiotics with other medicines or foods, which indicates a more relaxed attitude to antibiotics. Furthermore, Greek citizens seemed to be more knowledgeable about the proper use of antibiotics than were Turkish citizens.

### 3.5. Resistance and Prevention Measures

Participants were asked questions about the development of antibiotic resistance (13 questions, Table 5). Most participants were aware of the risk of antimicrobial resistance (Question 1; 77.6% in Greece and 80.7% in Turkey answered correctly). The risk of the unnecessary use of antibiotics (Question 7; 92.1% in Greece and 87.5% in Turkey) and the need to complete a course of antibiotics, even if feeling better (Question 8; 88.3% in Greece and 81.0% in Turkey), were also widely known, although this knowledge was higher among Greek people than in Turkish people. The most common misunderstandings were related to humans (instead of bacteria) becoming resistant (question 3; 60.4% of Greek and 76.2% of Turkish citizens answered “yes”, the country difference being statistically significant, *p* < 0.001). Greek respondents acknowledged, more often than Turkish respondents, that a person could be a “carrier” of resistant bacteria without getting ill. In addition, 16.9% of Greeks and 17.8% of Turks thought that resistance could spread from one person to another. Greek respondents agreed more than Turkish respondents with the statement that the more we use antibiotics in society, the higher the risk of resistance developing (*p* < 0.001). In general, Greek individuals were more knowledgeable about antibiotic resistance than Turkish individuals.

### 3.6. Attitudes to Antibiotic Use

The assessment of attitudes towards antibiotic use was conducted through ten statements, with responses recorded on a five-point Likert scale, which ranged from “strongly disagree” (1) to “strongly agree” (5). Table 6 presents the calculated means (M) and standard deviations (SD) for each sample, along with the results of the pairwise comparisons conducted through *t*-tests. Greek respondents agreed more than Turkish respondents with statement 5, while the Turkish individuals had higher average agreement scores with statements 1–4. There was no difference between Greek citizens and Turkish citizens in statements 6–10. Items 2, 3, 5, 8, 9 and 10 reflect a lenient attitude to antibiotic use, while items 1, 4, 6 and 7 indicate a preference for stricter control of antibiotics. In the independent samples *t*-test, no statistically significant difference was found for sum score “positive attitudes to antibiotic use”, t(705) = −1.43, *p* = 0.154. Overall, the two countries did not seem to greatly differ in their attitudes to antibiotics.

## 4. Discussion

The unnecessary use and overuse of antibiotics are the most important causes of antibiotic resistance in the community, which is mainly driven by the lack of knowledge and awareness regarding antibiotic use and the development of antimicrobial resistance. In the current study, the knowledge, attitudes and behaviours regarding antibiotic use and antibiotic resistance were investigated in two OECD countries with very high antimicrobial consumption and resistance rates, Turkey and Greece, in order to identify problematic attitudes, beliefs and behaviours and to guide future interventions to improve awareness and to reduce the misuse of antibiotics in both countries.

As reported in the OECD Health Policy Studies of 2018, antimicrobial resistance (AMR) rates have shown a persistent increase across all OECD countries from 2005 to 2015. The highest rates of AMR, approximately 35% in Turkey, Korea and Greece, were seven times higher than the lowest rates observed among OECD member countries. The projections made by the OECD suggest that AMR will continue to rise in OECD countries, placing a significant burden on the population’s health and on healthcare budgets. In response to this concerning trend, Turkey has implemented two main antimicrobial stewardship programs (ASPs), which were established by the Ministry of Health (MoH). The first ASP targeted hospitals, while the second focused on the community. The MoH initiated the second ASP, a 4-year National Action Plan for Rational Drug Use (2014–2017), after the World Health Organization (WHO) expressed concerns about the exceptionally high outpatient use of antibiotics. This community-level initiative aimed to raise public awareness of prudent antibiotic use, leading to public campaigns to discourage inappropriate antibiotic use. In 2015, over-the-counter (OTC) sales of antibiotics were prohibited (i.e., without a prescription). These efforts yielded positive results, reducing first-line prescription rates from 35% to 25% and dropping antimicrobial consumption of J01 class antibiotics (including b-lactams, tetracyclines, amphenicols, sulphonamides and trimethoprim, macrolides, aminoglycosides, quinolones) nationwide from 42.2 DID to 35.25 DID between 2011 and 2017. However, despite these improvements, antibiotic resistance remains a significant issue in Turkey. Likewise, Greece demonstrates a significant level of antibiotic consumption, ranking among the highest in Europe for both hospitals and the community, according to the European Centre for Disease Prevention and Control (ECDC) antimicrobial consumption database (ESAC-Net) [17]. The country also faces a correspondingly high prevalence of antimicrobial resistance [17]. Overall, both Turkey and Greece are grappling with the challenges posed by increasing AMR rates, necessitating further efforts to combat this issue effectively.

Based on the findings from the 2018 OECD Health Policy Studies, antimicrobial resistance (AMR) rates have exhibited a consistent and alarming rise in all OECD countries from 2005 to 2015. Notably, Turkey, Korea and Greece recorded the highest rates of antimicrobial resistance, reaching approximately 35%, which were seven times higher than the lowest rates observed among other member countries. These results indicate an urgent need to address the growing issue of AMR in OECD nations, as projected by the OECD, which predicts a continual increase in AMR rates, posing substantial challenges to population health and healthcare budgets [18].

In the current study, citizens of Turkey and Greece were investigated in terms of their experiences with antibiotics. In both communities, previous antibiotic use during the past year was over 40%. However, the vast majority of the citizens indicated that they followed doctor’s orders on antibiotic use and did not request antibiotics if they were not necessary. In terms of accessibility, 15.6% of the participants in Greece stated that it was difficult or extremely difficult to obtain antibiotics without a prescription, while this rate was much higher (40.6%) in Turkey. These results suggest a tighter control of OTC sales of antibiotics in Turkey compared to Greece. Self-medication has been previously reported in similar studies in both urban and rural areas in Greece, which showed the major source being the acquisition from pharmacies without prescription and the use of left-over antibiotics at home or, to a lesser extent, from friends/relatives [19,20,21]. Illegal access to antibiotics without a prescription has also been described in other European countries, such as Spain and Italy, where self-medication is also widespread [22,23,24]. The potential determinants of pharmacists dispensing antibiotics without a prescription (DAwMP) have been found to be due to the fear of losing a regular customer, knowing that the patient has difficulty in gaining access to a doctor (particularly in rural areas), insufficient knowledge (e.g., not recognizing antibiotic resistance as a healthcare problem) and the belief that new antibiotics will be discovered. In addition to self-medication, over-prescription by Greek physicians has been reported to be a major driver of antimicrobial overuse [25], although a carbapenem-focused antimicrobial stewardship programme has recently been implemented in the adult clinics of some hospitals in Greece [26].

For antibiotic stewardship to be effective at the community level, citizens must firstly have knowledge about drugs, as well as being able to recognize the types and classes of drugs they use. The results obtained in our study showed that the recognition scores in the community were higher for amoxicillin and tetracycline in Greece compared to those in Turkey. Importantly, more Turkish respondents thought that ibuprofen (12.8%) was an antibiotic, compared to Greek respondents (6.8%). Overall, the Greek citizens were found to be more knowledgeable about drug classes and were better able to recognize drugs than the Turkish respondents were. In a study performed on patients attending a hospital in the Marmara region of Turkey, the authors found that the recognition scores for antibiotics were even lower than those found in the current study. They found that 22% and 17% of the participants thought paracetamol and ibuprofen were antibiotics, respectively [27]. Among other countries in Europe, similar investigations have reported higher recognitions scores for antibiotics, such as a survey of parents in Italy in which 94% of the individuals correctly reported that paracetamol was not an antibiotic [28] and another survey where 87.1% of the respondents were aware that paracetamol was not an antibiotic [29]. In terms of the knowledge of antibiotic effectiveness, the current study found that although more Turkish citizens knew that antibiotics were effective against bacteria (87.2%, Greek citizens = 72.3), they also more often reported antibiotics to be effective against a cold or flu (27.1%) compared to the Greek participants (16.3%). This finding clearly indicates that the viral and self-limiting nature of the common cold is not a well-understood concept among both communities. Gaygısız et al. also reported a similar finding, with the authors revealing confusion about viral diseases in the general public of Turkey, which they suggested may explain the widespread use of antibiotics to self-treat seasonal influenzas and the common cold [13]. A previous study on the knowledge, attitudes and practices (KAP) of parents in Greece regarding antibiotic use for upper respiratory tract infections reported that although parents were aware of the fact that these infections are self-limiting, surprisingly, 74% expected to receive antibiotics when such a diagnosis was given [30]. When the side-effects of antibiotics were evaluated in the two communities, Turkish participants were less knowledgeable than Greek participants were, and they perceived the associated risks of taking antibiotics to be lower. However, the knowledge scores of the Turkish participants were higher overall than those found in a study performed in Turkey in 2019, in which only 31.3% of the citizens indicated that they knew about the side effects of antibiotics [31].

Public awareness of antibiotic resistance is a critical step in the prevention of AMR, which is a major threat to the clinical efficacy of antimicrobial medication. In the current study, most participants in both Turkey (80.7%) and Greece (77.6%) were aware that bacteria can become resistant to antibiotics. The vast majority of individuals from both communities were also aware of the risks of unnecessary antibiotic use (Turkey: 87.5%; Greece: 92.1%) and the need to fully complete a course of antibiotics (Turkey: 81.0%; Greece: 88.3%). A recent public survey of knowledge and attitudes regarding antibiotic use in the capital city Ankara (Turkey) found that while 67.5% of the participants were aware of the term antibiotic resistance, 17.1% of them could not link antibiotic use to the development of resistance [12]. A separate Turkish study reported that 28% of parents with children under the age of 18 did not think that the inappropriate use of antibiotics would change the effectiveness of the treatment and lead to an increase in bacterial resistance [32]. Similar to our study, a recent study of KAP on antibiotic use, performed during the COVID-19 pandemic in Greece, showed that 81% of the parents were aware of antimicrobial resistance developing due to antibiotic misuse [15]. Of note, the most common misunderstanding regarding antimicrobial resistance was that a large number of participants in both communities thought that humans could become resistant, rather than bacteria. When the attitudes to antibiotic use were evaluated in Turkey and Greece, Greek respondents were more prone to use leftover antibiotics or to give them to someone else later, although overall participants from the two countries did not differ greatly. Self-medication with left-over medications at home or obtained from pharmacies in Greece has also been previously described [20].

Turkey and Greece represent two countries that have been under the radar for antibiotic consumption and resistance rates, and ASPs have been implemented in both countries in an effort to curb the AMR problem. The findings of the current study indicate that the general public in Greece has a better recognition of antimicrobial drugs, knowledge of antibiotics and antibiotic resistance in comparison to that in Turkey, yet no differences were observed in terms of the attitudes towards antibiotic use. A multifaceted regional campaign in Greece, promoting prudent use of antibiotics, was found to lead to a rationalization in the choice of antimicrobials in both the general public and primary care physicians, but there was no reduction in the total antimicrobial consumption [33]. Interventions that aim to discourage unnecessary antibiotic use have also been implemented in Turkey, with the support and guidance of the WHO, and have been shown to be successful [34]. However, continued public education campaigns are required in both countries for a universal effort against AMR.

### 4.1. Study Strengths and Limitations

To the best of our knowledge, this is the first study to compare the KAP of citizens in Greece and Turkey, which have the leading antibiotic consumption and resistance rates in the Europe region. This study revealed important information on the knowledge gaps and problematic attitudes in both countries. However, the present study also has several limitations, the first of which is the relatively low number of participants from both countries. A second limitation could be that the majority of the participants in both communities had higher degrees, and whether participants were living in residential or rural areas was not measured. In addition, the samples were selected using social media advertising and snowball sampling, and so are not likely to be representative of the general populations of the two countries.

### 4.2. Future Perspectives and Recommendations

Continued interventions targeting the general population in both countries, such as public health education campaigns, are needed to encourage appropriate antibiotics use in the community. A stricter control of OTC sales of antibiotics through the enforcement of existing laws, particularly in rural areas, is also needed in both countries. Strategies such as the retention of prescriptions for antibiotics by pharmacies, government inspections and engaging pharmacists in the design of interventions should be considered. An integrated health strategy in both Turkey and Greece may help to reduce the uncontrolled antibiotic use, not only for human use but in all sectors involved.

## Figures and Tables

**Table 1 antibiotics-12-01279-t001:** Sample characteristics.

	Greece		Turkey		χ^2^
Variable	N	%	N	%	
Sex					*p* < 0.001
Female	152	49.4	249	62.4	
Male	151	49.0	147	36.8	
Missing	5	1.6	3	0.8	
“What is your marital status?”					*p* < 0.001
Married/Cohabiting	111	36.0	224	56.1	
Single	109	35.4	115	28.8	
In relationship	73	23.7	47	11.8	
Other	13	4.2	13	3.3	
Missing	2	0.6	0	0	
“What is your highest educational qualification?”					*p* < 0.001
Primary school/lower secondary school	3	1.0	5	1.3	
Upper secondary school/high school	48	15.6	20	5.0	
University/College ≤ 4 years	113	36.7	96	24.1	
University/College > 4 years	143	46.4	278	69.7	
Missing	1	0.3	0	0	
“Do you have a health professional background?”					*p* < 0.01
Yes	60	19.5	46	11.5	
No	239	77.6	352	88.2	
Missing	9	2.9	1	0.3	
“Do you smoke?”					*p* = NS
Yes	71	23.1	87	21.8	
No	202	65.6	275	68.9	
Sometimes	31	10.1	37	9.3	
Missing	4	1.3	0	0	
“I keep myself physically fit”					*p* < 0.05
Strongly agree	58	18.8	54	13.5	
Agree	120	39.0	165	41.4	
Neither agree nor disagree	93	30.2	109	27.3	
Disagree	27	8.8	64	16.0	
Strongly disagree	8	2.6	7	1.8	
Missing	2	0.6	0	0	
“I am in good health”					*p* = NS
Strongly agree	54	17.6	54	13.5	
Agree	158	51.5	216	54.1	
Neither agree nor disagree	83	27.0	102	25.6	
Disagree	10	3.3	25	6.3	
Strongly disagree	2	0.7	2	0.5	
Missing	1	0.3	0	0	

Notes: N denotes count; NS denotes “not significant” (*p* > 0.05); χ^2^ denotes Chi-Square Test of Independence.

**Table 2 antibiotics-12-01279-t002:** Antibiotic use and perceived availability in Greece and Turkey.

	Greece		Turkey		χ^2^
Variable	N	%	N	%	
“Have you taken antibiotics in the last 12 months?”					*p* = NS
Yes	138	44.8	169	42.4	
No	166	53.9	224	56.1	
Don’t know	2	0.6	6	1.5	
Missing	2	0.6	0	0.0	
“How many times have you taken a course of antibiotics during the past 12 months?”					*p* < 0.05
Never	164	53.2	229	57.4	
Once	96	31.2	98	24.6	
2–5 times	46	14.9	59	14.8	
More than 5 times	2	0.6	13	3.3	
“The last time you used antibiotics, did you follow the doctor’s instructions on dosage and length of treatment?”					*p* = NS
Yes	288	93.5	367	92.0	
No	17	5.5	23	5.8	
Don’t know	3	1.0	9	2.3	
“Have you ever asked a doctor for antibiotics, although the doctor deemed it unnecessary?”					*p* = NS
Yes, and received for myself	18	5.8	22	5.5	
Yes, and received for my child	1	0.3	3	0.8	
Yes, but did not receive	15	4.9	18	4.5	
No	272	88.3	356	89.2	
Missing	2	0.6	0	0.0	
“How easy to obtain antibiotics without a prescription?”					*p* < 0.001
Extremely easy	62	20.1	54	13.5	
Easy	100	32.5	88	22.1	
Neither easy nor difficult	95	30.8	95	23.8	
Difficult	40	13.0	132	33.1	
Extremely difficult	8	2.6	30	7.5	
Missing	3	1.0	0	0.0	

Notes: N denotes count; NS denotes “not significant” (*p* > 0.05); χ^2^ denotes Chi-Square Test of Independence.

**Table 3 antibiotics-12-01279-t003:** Recognition of antibiotics and a community comparison (Mann–Whitney U statistics).

	Greece (GR)			Turkey (TR)			Mann–Whitney U	Difference
Variable	No %	Don’t Know%	Yes%	No %	Don’t Know%	Yes%		
Tetracycline	2.6	51.0	44.5	2.5	70.9	26.6	*p* < 0.001	GR > TR
Penicillin	18.5	18.2	61.0	15.0	24.1	60.9	*p* = NS	
Ibuprofen	50.3	35.1	6.8	47.4	39.8	12.8	*p* < 0.05	TR > GR
Paracetamol	67.9	14.6	10.1	63.9	29.3	6.8	*p* < 0.05	TR > GR
Amoxicillin	2.6	25.6	70.1	1.0	53.9	45.1	*p* < 0.001	GR > TR

Notes: NS denotes not significant (*p* > 0.05).

**Table 4 antibiotics-12-01279-t004:** Knowledge about antibiotic use and the effects among the Greek and Turkish populations.

	Greece (GR)			Turkey (TR)			Mann–Whitney U	Difference
Variable	No %	Don’t know%	Yes%	No %	Don’t know%	Yes%		
1. Antibiotics kill viruses.	59.4	11.4	29.2	68.9	13.3	17.8	*p* < 0.01	GR > TR
2. Antibiotics are effective against bacteria.	10.4	17.3	72.3	5.8	7.0	87.2	*p* < 0.001	TR > GR
3. Antibiotics are effective against colds.	73.9	9.8	16.3	63.7	9.3	27.1	*p* < 0.001	TR > GR
4. Antibiotics are effective against flu.	60.0	12.5	27.5	61.2	6.8	32.1	*p* = NS	-
5. Penicillin is another word for antibiotics.	64.7	24.8	10.5	43.4	28.3	28.3	*p* < 0.001	TR > GR
6. One can get well from a bacterial infection without the use of antibiotics.	22.9	32.0	45.1	29.8	18.3	51.9	*p* = NS	-
7. Antibiotics may also kill beneficial bacteria that we normally carry on our skin or in our stomach/intestines.	4.9	17.2	77.9	6.5	14.5	78.9	*p* = NS	-
8. Antibiotics treatment often cause side-effects like diarrhoea.	8.1	25.3	66.6	12.8	27.3	59.9	*p* < 0.05	GR > TR
9. One side effect from antibiotics treatment is vaginal fungus infection in women.	4.2	27.6	68.2	15.8	40.6	43.6	*p* < 0.001	GR > TR
10. Antibiotics may influence the effect of other medications.	5.2	23.7	71.1	6.3	19.5	74.2	*p* = NS	-
11. Other medications may influence the effect of antibiotics.	2.6	20.1	77.3	8.5	22.3	69.2	*p* < 0.01	GR > TR
12. For some antibiotics combination with alcohol can be dangerous.	2.6	11.1	86.3	3.5	9.8	86.7	*p* = NS	-
13. Antibiotics can be used together with all kinds of food.	69.0	38.0	23.1	52.4	28.6	19.0	*p* < 0.01	GR > TR

Notes: NS denotes not significant (*p* > 0.05).

**Table 5 antibiotics-12-01279-t005:** Knowledge about antibiotic resistance and its effects among Greek and Turkish participants.

	Greece (GR)			Turkey (TR)			Mann–Whitney U	Difference
Variable	No %	Don’t Know%	Yes%	No %	Don’t Know%	Yes%		
1. Bacteria may become resistant toward antibiotics.	7.8	14.3	77.6	3.5	15.8	80.7	*p* = NS	-
2. Viruses may become resistant to antibiotics.	38.0	19.2	42.5	35.3	30.6	34.1	*p* = NS	-
3. Humans may become resistant to antibiotics.	21.4	17.5	60.4	8.0	15.8	76.2	*p* < 0.001	TR > GR
4. One can be a “carrier” of resistant bacteria, which means to have resistant bacteria in the body without being ill.	5.2	33.8	60.4	6.0	44.1	49.9	*p* < 0.01	GR > TR
5. Infections by resistant bacteria is increasing in Greece/Turkey.	2.9	42.5	53.9	1.0	42.6	56.4	*p* = NS	-
6. Resistant bacteria is a problem in Greek/Turkish hospitals.	3.9	32.8	63.0	2.8	46.6	50.6	*p* < 0.01	GR > TR
7. Unnecessary use of antibiotics can make them less effective.	2.6	5.2	92.2	3.8	8.8	87.5	*p* < 0.05	GR > TR
8. If you feel well halfway through the treatment that the doctor ordered, you can stop the antibiotic treatment.	88.3	6.2	5.2	81.0	10.5	8.5	*p* < 0.01	TR > GR
9. Greeks/Turks can help to prevent antibiotic resistance.	7.1	31.8	60.7	6.8	25.3	67.9	*p* = NS	-
10. Frequent use of antibiotics on animals can reduce the effectiveness of antibiotics in humans.	23.1	46.4	30.2	23.6	46.4	30.1	*p* = NS	-
11. Antibiotic resistance can spread from person to person.	54.5	27.6	16.9	40.9	41.4	17.8	*p* < 0.01	TR > GR
12. Antibiotic resistance can spread from animals to humans	39.9	40.3	19.2	33.3	48.4	18.3	*p* = NS	-
13. The more antibiotics we use in society, the higher the risk that resistance develops and spreads.	8.1	14.9	76.6	17.8	32.8	49.4	*p* < 0.001	GR > TR

Notes: NS denotes not significant (*p* > 0.05).

**Table 6 antibiotics-12-01279-t006:** Attitudes to antibiotic use among the Greek and Turkish populations.

	Greece		Turkey		
Variable	M	SD	M	SD	t
1. I want to use antibiotics only if it is necessary.	4.70	0.54	4.81	0.55	*p* < 0.01
2. I want to use antibiotics if it makes me get well sooner.	3.41	1.16	3.61	1.26	*p* < 0.05
3. The doctor should give me antibiotics when I think I need it.	2.23	1.40	3.58	1.47	*p* < 0.001
4. The doctor should not give me antibiotics when he/she thinks I do not need it.	4.38	1.08	4.65	0.82	*p* < 0.001
5. Leftover antibiotics can be saved for personal use in the future or given to someone else.	2.90	1.24	2.32	1.24	*p* < 0.001
6. Leftover antibiotics should be taken back to the pharmacy.	3.25	1.15	3.29	1.24	*p* = NS
7. It is good that one needs a prescription to acquire antibiotics from pharmacies in Greece/Turkey.	4.37	0.83	4.41	1.01	*p* = NS
8. It is good to be able to buy antibiotics online, without having to see a doctor.	1.77	1.04	1.91	1.19	*p* = NS
9. It is good to be able to acquire antibiotics from relatives or acquaintances, without having to be examined by a doctor.	1.71	0.96	1.85	1.18	*p* = NS
10. It is good that one can buy antibiotics without a prescription in pharmacies in Greece/Turkey.	1.99	1.07	1.85	1.18	*p* = NS

Notes: M denotes mean; SD denotes standard deviation; NS denotes not significant (*p* > 0.05); t denotes Student’s *t*-test of independent means.

## Data Availability

The datasets generated and analysed during the current study are not publicly available due to the fact that the datasets include multiple extensive files with a large number of variables, but they are available from the corresponding author upon request.

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
