# Peer review of "Antibiotics Knowledge, Attitudes and Behaviours among the Population Living in Greece and Turkey"

_antibiotics, 2023, doi:10.3390/antibiotics12081279_

Round 1

Reviewer 1 Report

Thank you for involving me in reviewing this manuscript entitled “Antibiotics knowledge, attitudes and behaviours among population living in Greece and Turkey” which investigated the knowledge and attitudes towards antibiotics in the public living in Greece and Turkey.

Some comments were raised and must be addressed before phasing up.

1.      Abstract: the authors did not write anything about the methods used in the study.

In addition, this section must be in structured format.

Please add numbers to the results.

This “This study highlights the fact that different countries have varying levels of 25 knowledge, attitudes and perceptions regarding antibiotic use and antibiotic resistance.” Is unclear! And I don’t believe the comparison was the aim of the study.

Introduction

This section is relatively long, and I suggest keeping it more focused and to the point!

Furthermore, with the listed studies conducted in Turkey, the research gap is unclear!

Methods

1.      This section must be transferred to after introduction, following the authors guidelines.

2.      This section must include many more data regarding:

Tool development and validation, including face and content validity.

Sample size calculations

Inclusion and exclusion criteria of the participants.

Administration language ? translation and back-translation.

Discussion

Line 237 What do you mean by “A part of the programme aimed to raise public awareness of prudent 237 antibiotic use and public campaigns were initiated to discourage inappropriate antibiotic 238 use, and over-the-counter (OTC) sales of antibiotics were prohibited in 2015.” ? was it allowed to sale the antibiotics as OTC medication?  This sentence is tricky, please clarify.

Line 248: why Antimicrobial resistance in Greece is similarly high ? Elaborate please.

Study limitations are much more the mentioned ones: selection bias, use of snowballing sampling, online survey….etc

Author Response

We would like to thank you for your comments, which helped us to improve the manuscript and to correct some mistakes. We have addressed all your revisions requests.

 “ Abstract: the authors did not write anything about the methods used in the study.”

We have added the main details about the methods used.

"In addition, this section must be in structured format.”

The abstract is in the structured format required, the only missing aspect was the method.

“Please add numbers to the results.”

We have added numbers (% and p-value) when possible

“This “This study highlights the fact that different countries have varying levels of knowledge, attitudes and perceptions regarding antibiotic use and antibiotic resistance.” Is unclear! And I don’t believe the comparison was the aim of the study.”

Thank you for this comment. The aim was to compare Greece and Turkey (two high antibiotic use countries) in above-mentioned aspects. We have rewritten the sentence to cover only these two countries.

“Introduction”

"This section is relatively long, and I suggest keeping it more focused and to the point!"

We have cut the introduction back considerably.

“Furthermore, with the listed studies conducted in Turkey, the research gap is unclear!”

The gaps are firstly that only one study has investigated more than one city in Turkey and that was based upon university students. Secondly, this study is the first to compare the attitudes, knowledges and practices of Greece and Turkey. We have tried to make these gaps more obvious, without greatly adding to the length of the introduction.

“Methods”

“This section must be transferred to after introduction, following the authors guidelines.”

The method section has been moved to be between the introduction and the results sections, as suggested.

"This section must include many more data regarding: Tool development and validation, including face and content validity. Sample size calculations. Inclusion and exclusion criteria of the participants.”

These all are now included in the method section. The questions have been validated in an earlier study (Sullman, M.; Lajunen, T.; Baddal, B.; Apostolou, M. Antibiotics Knowledge, Attitudes and Behaviours among the Population Living in Cyprus. Antibiotics 2023, 12(5), 897; https://doi.org/10.3390/antibiotics12050897) and been used in other previous studies cited in the text.

“Administration language ? translation and back-translation.

This has now been added to the method section (Greek and Turkish).

“Discussion”

"Line 237 What do you mean by “A part of the programme aimed to raise public awareness of prudent 237 antibiotic use and public campaigns were initiated to discourage inappropriate antibiotic 238 use, and over-the-counter (OTC) sales of antibiotics were prohibited in 2015.” ? was it allowed to sale the antibiotics as OTC medication?  This sentence is tricky, please clarify."

This is a part of the National Action Plan for Rational Drug Use that is mentioned in the previous sentence. OTC were prohibited in 2015, mean that is it not allowed to sell antibiotics OTC (i.e., without a prescription). We are unsure what is tricky about this sentence, but have attempted to improve the clarity. 

"Line 248: why Antimicrobial resistance in Greece is similarly high ? Elaborate please."

The main reasons are already stated in the first lines of the Discussion section “The unnecessary use and overuse of antibiotics are the most important causes of antibiotic resistance in the community”. This sentence was not well worded and possibly the reviewer missed this sentence due to that, so we have fixed the English.

 "Study limitations are much more the mentioned ones: selection bias, use of snowballing sampling, online survey….etc"

We have added a separate strengths and limitation section and added extra material to this section. The main limitation is that the use of Facebook advertising and snowball sampling means it is unlikely that the samples are representative of the two populations. This was added to as a limitation. However, we do not agree that having an online survey is a limitation. There are positive and negative aspects to the use of an online survey, which is beyond the scope of this article to discuss.

Reviewer 2 Report

The article “Antibiotics knowledge, attitudes and behaviours among population living in Greece and Turkey” is very interesting but I have following comments/suggestions,

1.       How was the sample size calculated? Which method was used?

2.       Can you explain how the snowball sampling was used?

3.       How was the response rate calculated? If not explain why.

4.       Which guidelines were used to report this online study? (Kindly refer to CHERRIES)

5.       How was the reliability of the questionnaire assessed?

6.       What was the internal consistency of the tool?

7.       What was the language of the study questionnaire?

8.       Table 1, how can the authors how SD or % in one column?

9.       Table 2, explain NS in the footnotes.

Author Response

We would like to thank you for your comments, which helped us to improve the manuscript and to correct some mistakes. We have addressed all your revisions requests.

  1. “How was the sample size calculated? Which method was used?”

We have added description of the sample size calculation to the method section.

  1. “Can you explain how the snowball sampling was used?”

We have added an additional sentence to explain the snowball sampling.

  1. “How was the response rate calculated? If not explain why.”

It is not possible to calculate a response rate, since we advertised on social media (e.g., Facebook), contacted colleagues and friends and asked them to pass on the link to their network, asked our local students to pass the link on to their friends and family. Thus, as is always the case with snowball sampling, it is impossible to calculate a response rate.

  1. “Which guidelines were used to report this online study? (Kindly refer to CHERRIES)”

Thank you, we forgot to mention this, but have now added it to the paper.

  1. “How was the reliability of the questionnaire assessed? What was the internal consistency of the tool?”

The study is based on analysis of individual items, not composite scores, because the analysis of individual items is more revealing than the sum scores. Moreover, all scales except attitudes were measured with response alternatives “yes”, “no” and "don’t know”. Since this level of measurement can be considered categorical, we had to use non-parametric statistics and could not calculate sum scores. Responses to the 10-item attitude scale was based on a 5-point likert scale, so we actually could calculate a sum score and reliability coefficient. These have been added to the method section and t-test about country differences was added to the results section. This comparison confirmed the results observed in Table 6.

  1. “What was the language of the study questionnaire?”

The local language was used (Greek for Greece and Turkish for Turkey). This is stated in the method section, and the translation process is also now described.

  1. “ Table 1, how can the authors how SD or % in one column?”

Thank you, this was a good observation. The only variable with M and SD was age, and we decided to report it in the text. Now the table has only N and %, which is clearer.

  1. Table 2, explain NS in the footnotes.

Done.

Reviewer 3 Report

Thanks for giving me the opportunity to review this manuscript.

The manuscript is good but I have suggestions that need to be resolved.

-in title author wrote knowledge, attitudes and behaviours but in abstract they mention that their aim is to check the knowledge, and attitudes. resolve it.

-add some key information about methods in abstract.

-add main results in the abstract along with the statistical test results such as p -value.

-in introduction section - mention the reference in the line 106 antibiotics among the .......two of the largest users of antibiotics in the European area.

-in results section - in table 1 mention which statistical test used in the footnote of the table. mention what is 2 in the header of table 1, and what is NS in table 1.

-in results section - in table 2 authors mention xtest used, add Chi-square in the footnote of the table. mention what is NS in table 2.

-in results section - in table 3 authors mention difference kindly mention how they calculate, mention what is NS in table 3. In the footnote, write *p<0.05; **p<0.01; ***p<0.001 

-why did the author not calculate the knowledge score as they did in attitude in Table 5? It must be required  

 -need to explain the method section in detail.

-in method section - mention different subheadings in this section, such as study participants, study design, questionnaire development, Analysis etc.

-add strengths and limitation of the study under seprate heading.

-add future perspective and recommendations

Author Response

We would like to thank you for your comments, which helped us to improve the manuscript and to correct some mistakes. We have addressed all your revisions requests.

“in title author wrote knowledge, attitudes and behaviours but in abstract they mention that their aim is to check the knowledge, and attitudes. resolve it.”

We have changed this to include behaviours, in the two areas we noted this was missing.

“add some key information about methods in abstract.”

We have added the most important information (cross-sectional online survey, using Facebook advertising and snowball sampling) in the abstract.

“add main results in the abstract along with the statistical test results such as p -value.”

We have added statistics to abstract whenever possible.

“in introduction section - mention the reference in the line 106 antibiotics among the .......two of the largest users of antibiotics in the European area.”

When cutting back the introduction, as per the suggestion of R1, we have removed this statement, since it is a repeat of the same statement earlier. We hope this solves the issue.

“in results section - in table 1 mention which statistical test used in the footnote of the table. mention what is 2 in the header of table 1, and what is NS in table 1.”

We have added this information.

“in results section - in table 2 authors mention xtest used, add Chi-square in the footnote of the table. mention what is NS in table 2.”

We have added this information

“in results section - in table 3 authors mention difference kindly mention how they calculate, mention what is NS in table 3. In the footnote, write *p<0.05; **p<0.01; ***p<0.001 “

We have revised the table. The name of the test (Mann–Whitney U statistics) and the p-values are given in the table.

“why did the author not calculate the knowledge score as they did in attitude in Table 5? It must be required “

Knowledge of antibiotics is a categorical measure (“no”, “yes”, “don’t know”) and, therefore, we had to use a non-parametric test. For the same reason, summing up the items is not preferable. Comparison of individual antibiotics knowledge is more informative since some of the antibiotics are very well known by public (penicillin) but other not (e.g. tetracycline). Attitude score in Table 5 is based on 5-point likerts scale which allows using parametric tests. It also allows summing up the item scores. We have added this to the paper, although the result just confirms what can be seen in the t-test of individual items.

“need to explain the method section in detail. in method section - mention different subheadings in this section, such as study participants, study design, questionnaire development, Analysis etc.”

We have revised the method section and added more information.

“add strengths and limitation of the study under seprate heading”.

We have added the strengths and limitation section and added extra material to this section.

-“add future perspective and recommendations”

We have added a paragraph on future perspectives and recommendations.

Round 2

Reviewer 1 Report

Thank you for addressing the comments, it looks better now! 

Reviewer 2 Report

The authors have addressed all of my comments/suggestions in their revised submission. 

Reviewer 3 Report

None